# Diagnostic Ability and Safety of Repeated Pancreatic Juice Cytology Using an Endoscopic Nasopancreatic Drainage Catheter for Pancreatic Ductal Adenocarcinoma: A Multicenter Prospective Study

**DOI:** 10.3390/diagnostics13162696

**Published:** 2023-08-17

**Authors:** Shinya Nakamura, Yasutaka Ishii, Masahiro Serikawa, Keiji Hanada, Noriaki Eguchi, Tamito Sasaki, Yoshifumi Fujimoto, Atsushi Yamaguchi, Shinichiro Sugiyama, Bunjiro Noma, Michihiro Kamigaki, Tomoyuki Minami, Akihito Okazaki, Masanobu Yukutake, Teruo Mouri, Yumiko Tatsukawa, Juri Ikemoto, Koji Arihiro, Shiro Oka

**Affiliations:** 1Department of Gastroenterology, Graduate School of Biomedical & Health Sciences, Hiroshima University, Hiroshima 734-8551, Japan; shin617@hiroshima-u.ac.jp (S.N.); serikawa@hiroshima-u.ac.jp (M.S.); ymkt@hiroshima-u.ac.jp (Y.T.); juri1118@hiroshima-u.ac.jp (J.I.); oka4683@hiroshima-u.ac.jp (S.O.); 2Department of Gastroenterology, Onomichi General Hospital, Onomichi 722-8508, Japan; kh-ajpbd@nifty.com; 3Department of Gastroenterology, Hiroshima Memorial Hospital, Hiroshima 730-0802, Japan; eguchi@kkrhiroshimakinen-hp.org; 4Department of Gastroenterology, Hiroshima Prefectural Hospital, Hiroshima 734-8530, Japan; tamito0611@gmail.com; 5Department of Gastroenterology, Hiroshima General Hospital, Hatsukaichi 738-8503, Japan; fujiyt@yc4.so-net.ne.jp; 6Department of Gastroenterology, National Hospital Organization Kure Medical Center and Chugoku Cancer Center, Kure 737-0023, Japan; yamaguchi.atsushi.uc@mail.hosp.go.jp; 7Department of Gastroenterology, Saiseikai Hiroshima Hospital, Aki 731-4311, Japan; sugishin111@gmail.com; 8Department of Gastroenterology, Kure Kyosai Hospital, Kure 737-8508, Japan; b-noma@kure-kyosai.jp; 9Department of Gastroenterology, Saiseikai Kure Hospital, Kure 737-0921, Japan; m_kamigaki05@yahoo.co.jp; 10Department of Gastroenterology, Hiroshima Red Cross & Atomic-Bomb Survivors Hospital, Hiroshima 730-8619, Japan; tomminami0805@outlook.com; 11Department of Gastroenterology, National Hospital Organization Higashihiroshima Medical Center, Higashihiroshima 739-0041, Japan; ak.from.coast@gmail.com; 12Department of Gastroenterology, Hiroshima City North Medical Center Asa Citizens Hospital, Hiroshima 731-0293, Japan; m_yukutak@yahoo.co.jp; 13Department of Gastroenterology, Chugoku Rosai Hospital, Kure 737-0193, Japan; teruo0914@chugokuh.johas.go.jp; 14Department of Anatomical Pathology, Hiroshima University Hospital, Hiroshima 734-8551, Japan; arihiro@hiroshima-u.ac.jp

**Keywords:** pancreatic ductal adenocarcinoma, pancreatic juice cytology, endoscopic nasopancreatic drainage, early diagnosis, pancreatitis

## Abstract

Pathological examination is essential for the diagnosis and treatment of pancreatic ductal adenocarcinoma (PDAC). Moreover, a reliable pathological diagnosis is extremely important for improving prognosis, especially in early-stage PDAC. This study prospectively evaluated the usefulness of repeated pancreatic juice cytology (PJC) using an endoscopic nasopancreatic drainage (ENPD) catheter for the diagnosis of PDAC. We enrolled 82 patients suspected of having resectable PDAC, based on imaging studies, and judged the necessity for cytology. The diagnostic yield of up to six repeated PJCs and the incidence of complications, such as pancreatitis, was evaluated. A total of 60 patients were diagnosed with PDAC. The overall sensitivity and specificity were 46.7% and 95.5%, respectively. The cumulative positivity rate increased with the number of sampling sessions, reaching 58.3% in the sixth session. The sensitivity was significantly higher in the pancreatic head than in the pancreatic tail (*p* = 0.043). Additionally, it was 100% in four patients with a tumor size ≤10 mm. Pancreatitis occurred in six patients (7.3%), all of whom were treated conservatively. In the diagnosis of PDAC, repeated PJC using an ENPD catheter revealed a cumulative effect of sensitivity up to six times and an excellent diagnostic yield for small PDAC.

## 1. Introduction

Despite recent advances in diagnostic imaging and treatment, pancreatic ductal adenocarcinoma (PDAC) has a poor prognosis, with an overall 5-year survival rate of approximately 10% [1,2,3,4]. In patients with suspected PDAC, first noninvasive imaging studies, such as computed tomography (CT) and magnetic resonance imaging (MRI), are performed, followed by endoscopic examinations, such as endoscopic ultrasonography (EUS) and endoscopic retrograde cholangiopancreatography (ERCP), as needed. However, approximately 10% of patients in whom PDAC was suspected and resected had a benign disease [5]. Furthermore, imaging alone has limitations in the diagnosis of PDAC. Therefore, it is important to obtain pathological evidence, as recommended by the Japanese clinical practice guidelines [6]. In recent years, the usefulness of neoadjuvant chemotherapy for PDAC has been reported [7,8] and the importance of pathological diagnosis in determining treatment strategies has increased.

The methods used to obtain a pathological diagnosis of PDAC include endoscopic ultrasound-guided fine-needle aspiration (EUS-FNA) and ERCP. EUS-FNA is the suggested method when a pancreatic mass is found, owing to its higher diagnostic performance and lower complication rate compared to ERCP [9,10]. The widespread use of EUS-FNA has enabled pathological diagnosis of many PDACs, and the opportunities for cytological diagnosis by ERCP have decreased significantly. However, the sensitivity of EUS-FNA is reportedly reduced in small malignant pancreatic tumors [11]. Additionally, EUS-FNA is challenging to perform in certain cases because of the use of oral antithrombotic agents, the tendency for bleeding, or blood vessels interfering with the puncture route. Although its frequency is low, needle tract seeding using transgastric EUS-FNA has been reported [12]. Pathological diagnostic methods performed under ERCP include cytology of the pancreatic juice aspirated with a cannula inserted into the main pancreatic duct (MPD), brush cytology, and cytology of the pancreatic juice collected using an endoscopic nasopancreatic drainage (ENPD) catheter. Additionally, pancreatic juice cytology (PJC) using an ENPD catheter is known as serial pancreatic juice cytologic examination (SPACE) [13], and may be collected multiple times by aspiration or spontaneous dripping. Reports indicate that repeated PJC via an ENPD catheter is useful for the diagnosis of early-stage PDAC [14,15,16]. However, no reports have prospectively examined the diagnostic performance and safety of repeated PJC via an ENPD catheter in patients with suspected PDAC on imaging studies. Furthermore, the cumulative effect of repeatedly submitting pancreatic juice for cytology has not been clarified.

This novel study aimed to prospectively investigate the diagnostic performance and safety of repeated PJC via an ENPD catheter in patients with suspected PDAC, and to clarify its role in the diagnosis of PDAC.

## 2. Materials and Methods

### 2.1. Study Design

This multicenter, open-label, uncontrolled study was designed to evaluate the diagnostic performance and safety of repeat PJC via an ENPD catheter in patients with suspected PDAC. This study was conducted at the Hiroshima University Hospital and 12 affiliated hospitals. The study protocol is taken from the University Hospital Medical Information Network Clinical Trials Registry (UMIN000022268).

This study was conducted in accordance with the Declaration of Helsinki and was approved by the Ethics Committee of Hiroshima University (approval number: C-84).

### 2.2. Patient Selection

Patients suspected of having resectable PDAC, based on imaging studies such as computed tomography (CT), magnetic resonance imaging (MRI), and endoscopic ultrasound (EUS), and who required cytology between May 2016 and December 2018 were enrolled. Written informed consent was obtained from all enrolled patients before ERCP.

Exclusion criteria included patients (1) under the age of 20, (2) with acute pancreatitis, (3) with a history of pancreatectomy, (4) with a history of gastrectomy (excluding Billroth I reconstruction), (5) clinically suspected of having intraductal pancreatic mucinous neoplasm (IPMN) with obvious mucus in the duct, (6) after duodenal main papilla procedures such as endoscopic sphincterotomy and endoscopic papillary balloon dilation, (7) who could not give informed consent, (8) with a history of allergy to drugs used in routine ERCP, such as pancreatic enzyme inhibitors, antibiotics, sedatives, antispasmodics, anesthetic agents, and contrast medium or polyurethane products, (9) deemed by the investigator to be unsuitable for this study (e.g., due to pregnancy).

Patients with successful ENPD catheter placement and at least one PJC submission via an ENPD catheter were included in the analysis.

### 2.3. Endoscopic Retrograde Cholangiopancreatography and Nasopancreatic Drainage Catheter Placement

After pancreatography, a 0.025-inch guidewire was inserted into the distal MPD. Brush cytology for MPD stenosis was accomplished, as deemed necessary, by the attending physician, and an ENPD catheter was subsequently placed over the guidewire. Either a 4-Fr or 5-Fr ENPD catheter (Gadelius Medical, Tokyo, Japan) made of polyurethane was used at the discretion of the attending physician. It had a looped shape from the papilla of Vater to the duodenal bulb, with a total length of 260 cm. The portion of the catheter inserted into the pancreas was 10 cm long, and the tip was usually located in the pancreatic body (Figure 1). All ERCP-related procedures were performed under conscious sedation with intravenous administration of midazolam, diazepam, and flunitrazepam, as well as pentazocine, if necessary.

### 2.4. Protocol

After placement of the ENPD catheter, pancreatic juice was collected for 15 min each time by spontaneous dripping and was promptly submitted for cytological examination. As a rule, the ENPD catheter was left in place for 48 h, and the pancreatic juice was collected up to six times. The upper limit of the six submissions was determined by considering the balance between the medical fee for cytology in Japan and the cost of specimen preparation. After the ENPD catheter was placed, clinical symptoms were carefully observed, and blood tests, such as complete blood count, hepatobiliary enzymes, pancreatic amylase, and C-reactive protein, were performed both 2–4 h later and the next day to assess the presence or absence of complications. The ENPD catheter was removed after the scheduled number of samples for cytological examination had been submitted, or when the attending physician deemed it difficult to continue catheter placement due to complications.

### 2.5. Cytological Diagnosis

Specimens were stained with Papanicolaou stain after smear preparation and were diagnosed by multiple pathologists and one cytopathologist. The suitability of the samples was evaluated based on the degree of denaturation and the sufficiency of sample volume to make a diagnosis. Cytological diagnoses were categorized into the following three groups: negative (normal or reactive process), atypical (atypical cells of unknown malignancy), and positive (strongly suspected or definite malignancy).

### 2.6. Endpoints and Definitions

The primary endpoint was the sensitivity of repeated PJC using an ENPD catheter. The correct diagnosis was defined as follows: (1) When the PJC was positive, the resected specimen was histopathologically diagnosed as PDAC; (2) When the PJC and other pathological examinations, such as EUS-FNA, were negative or suspicious, surgery was performed, as the PDAC could not be ruled out by imaging. As a result, PDAC was not found histopathologically in the resected specimen; (3) When observation was chosen because PJC and other pathological examinations were negative or atypical, PDAC was negative on imaging at six months.

The secondary endpoint was the incidence of complications associated with the placement and removal of the ENPD catheter. Acute pancreatitis, hyperamylasemia, and acute cholangitis were considered as complications. Patients with acute pancreatitis had symptoms (e.g., abdominal pain and back pain) and elevated serum pancreatic enzyme (amylase and/or pancreatic amylase) levels more than three times the upper limit of normal. The severity of pancreatitis was assessed according to the Japanese severity criteria [17]. Hyperamylasemia was defined as an elevation in serum amylase or pancreatic amylase levels three times higher than the upper limit of normal, without symptoms or imaging findings of acute pancreatitis. Cholangitis was defined as an elevated hepatobiliary enzyme with the onset of high fever (≥38 °C).

### 2.7. Statistical Analyses

Statistical analyses were performed using JMP Pro 16.0.0 (SAS Institute Inc., Cary, NC, USA). Continuous variables were compared using the Wilcoxon rank-sum test, and categorical values were compared using the chi-square test or Fisher’s exact test. Diagnostic abilities based on lesion location and tumor size were compared among the three groups using Bonferroni adjustment for multiple comparisons. The level of significance was set at *p* values < 0.05.

## 3. Results

### 3.1. Study Flow Chart

During the study period, 91 patients with suspected resectable PDAC indicated for PJC with an ENPD catheter were enrolled in this study. Of the 91 patients, one was transferred to another hospital before ERCP was performed. Among the 90 patients who underwent ERCP, pancreatography could not be performed in four patients, and the ENPD catheter could not be placed in four patients due to the morphology of the pancreatic duct and the difficulty associated with inserting the guidewire. Finally, the remaining 82 patients underwent at least one PJC by ENPD catheter and were included in the analysis of this study.

Ultimately, 60 patients (73.2%), including two patients with carcinoma in situ, were diagnosed with PDAC, 50 of whom underwent surgical resection. Twenty-two patients (26.8%) were diagnosed with diseases other than PDAC, eight of whom underwent surgical resection. A flowchart of the study is shown in Figure 2.

### 3.2. Patient Characteristics

The characteristics of the 82 patients are presented in Table 1. The median age was 72 years (interquartile range [IQR]: 64–79 years), and the patients included 50 men and 32 women. The indications for examination were a pancreatic mass in 60 patients (73.2%), MPD stenosis in 13 (15.9%), MPD dilatation in 6 (7.3%), and cystic lesions in 3 (3.7%). The diameter in the 58 patients with pancreatic masses was ≤10 mm in 9 (15.5%), 11–20 mm in 27 (46.6%), and ≥21 mm in 22 (37.9%). The lesions were located in the pancreatic head in 38 (46.3%), body in 25 (30.5%), and tail in 19 (23.2%) patients.

The ENPD catheters were 4-Fr in 62 patients (75.6%) and 5-Fr in 20 patients (24.4%). PJC via an ENPD catheter was performed with a mean and median of 4.6 and 5 times, respectively (IQR: 4–6). Surgical resection was performed in 58 (70.7%) patients. The final diagnosis was PDAC in 60 patients (73.1%), including two patients with carcinoma in situ, intraductal papillary mucinous neoplasm in two, chronic pancreatitis in 11 (13.4%), autoimmune pancreatitis in one, pancreatic cyst in five, and others in three. Of 60 patients with PDAC, 50 (83.3%) underwent surgical resection. Of the 10 patients who did not undergo surgical resection, six were diagnosed as unresectable, owing to distant metastasis on additional imaging studies such as MRI and positron emission tomography/computed tomography, one ended in exploratory laparotomy, one was declared inoperable, and two did not undergo surgery, at their request. None of the 14 patients initially diagnosed with the benign disease was diagnosed with PDAC during the 6-month follow-up period.

### 3.3. Diagnostic Ability of Pancreatic Juice Cytology via an ENPD Catheter

The rate of adequate sample collection was 78.6% (287/365). Table 2 shows the final diagnosis, based on histopathological examination of specimens obtained by surgical resection or EUS-FNA, or clinical follow-up for each cytological diagnosis. Twenty-nine patients (35.4%) had positive PJC, and 28 patients (96.6%) were diagnosed with PDAC. Twenty-three patients (28.0%) had atypical PJC, and 17 patients (73.9%) were diagnosed with PDAC. Thirty patients (36.6%) had negative PJC, and 15 patients (50.0%) were diagnosed with PDAC. The overall sensitivity, specificity, positive predictive value, negative predictive value, and accuracy of the PJC via an ENPD catheter for PDAC were 46.7% (28/60), 95.5% (21/22), 96.6% (28/29), 39.6% (21/53), and 59.8% (49/82), respectively (Table 3). The positivity rates for each session of PJC, from one to six times, were 31.7% (19/60), 25.9% (14/54), 20.0% (10/50), 35.0% (14/40), 37.1% (13/35), and 45.8% (11/24), respectively (Table 4). The cumulative positive rate in the 24 patients with PDAC who underwent six PJC sessions is disclosed in Figure 3. The cumulative positive rate remained flat at 37.5% from the 1st to the 3rd sessions; however, it subsequently increased with the number of sampling sessions, reaching 58.3% at the 6th session.

The diagnostic yields based on tumor factors and diagnostic procedures are exhibited in Table 5. By the location of the tumor, the sensitivity of repeated PJC via an ENPD catheter was 58.6% (17/29), 52.6% (10/19), and 16.7% (2/12) for the pancreatic head, body, and tail, respectively. There was a significant difference among the three groups (*p* = 0.045), and sensitivity was significantly higher in the pancreatic head than in the pancreatic tail (*p* = 0.043). PDAC size was measured using resected specimens from resected patients and CT scans from non-resected patients. There was no significant difference (*p* = 0.073) in tumor size among the three groups (≤10 mm, 11–20 mm, and >20 mm); however, the sensitivity was 100% in four patients with a tumor size of 10 mm or less. All lesions in the four patients with PDAC of 10 mm were located in the pancreatic body. The sensitivity was significantly higher in patients with MPD stenosis on imaging studies such as CT, MRI, and EUS than in those without MPD stenosis. No significant difference was noted in the sensitivity of ENPD catheter diameter (4-Fr vs. 5-Fr) and brush cytology (*p* = 0.500 and 0.511, respectively).

### 3.4. Complications

The complications are displayed in Table 6. Pancreatitis occurred in six patients (7.3%) and was classified according to severity as mild, moderate, or severe. One patient was diagnosed with mild pancreatitis after ENPD catheter removal. All patients were treated conservatively. Hyperamylasemia not diagnosed as pancreatitis was present in 44 patients (53.7%), none of whom developed cholangitis. No relationship was found between catheter diameter and the incidence of each complication.

## 4. Discussion

EUS-FNA is the first choice for the pathological diagnosis of PDAC, owing to its excellent diagnostic performance and safety. PJC using ERCP is the gold standard pathological diagnostic method in cases where EUS-FNA is challenging to perform; however, its diagnostic yield is not satisfactory [18,19,20,21]. In this study, the sensitivity of a single PJC was low (31.7%); however, repeated PJC using an ENPD catheter resulted in a positive rate of 58.3% at the 6th session due to the cumulative effect in patients who underwent six sessions of PJC. No studies have reported the optimal number of PJC using an ENPD catheter; however, two studies have reported the optimal number of samples for bile aspiration cytology [22,23]. Both studies reported that the optimal number of samples was six for patients who underwent bile aspiration cytology via endoscopic nasobiliary drainage or percutaneous transhepatic biliary drainage. Here, the cumulative effect of PJC up to six times was confirmed; however, it is not possible to determine whether six samples are optimal for the diagnosis of PDAC, as in the above studies, unless the number of samples is increased further. However, submission of a PJC for seven or more sessions is challenging because of the discomfort of the patient associated with the prolonged ENPD catheter placement period and medical economics; therefore, six sessions may be considered the optimal number of repeated PJC using an ENPD catheter.

In this study, there was a characteristic relationship between tumor location and size and the diagnostic performance of repeat PJC. Regarding tumor location, sensitivity was significantly higher in the pancreatic head than in the pancreatic tail. Mikata et al. [15] stated that the sensitivity of repeated PJC via the ENPD catheter was better in the pancreatic head than in the pancreatic body and tail (90%, 69%, and 67%, respectively). Additionally, they suggested that this was because the tip of the ENPD catheter was located upstream of the stricture in the pancreatic head lesions, resulting in an increase in exfoliated cells by mechanical irritation from the ENPD catheter. However, the tip of the ENPD catheter used in this study was located downstream of the lesion in the PDAC of the pancreatic tail, which may have resulted in fewer exfoliated cells and lower sensitivity. Regarding the tumor diameter, the smaller the tumor, the higher the sensitivity. The sensitivity was 100%, especially in patients with PDAC of 10 mm or less. Similarly, several reports establish the usefulness of repeated PJC using an ENPD catheter in the diagnosis of patients with small PDAC [14,15,16,24], and the diagnostic sensitivity of carcinoma in situ has been reported to be 72.2–100%, which is extremely good. One of the reasons PJC has good sensitivity in small PDAC is the high frequency of intraductal spread along the main pancreatic duct [25,26]. Additionally, Yokode et al. [27] reported that, in ten cases of high-grade PanIN, lesions were chiefly present in the main pancreatic duct, with a median longitudinal extension of 18 mm. The prognosis of PDAC is very poor; however, the 5-year survival rates for Union for International Cancer Control stage 0 (in situ) cases and cases with tumors with a diameter of 10 mm or less, which account for most stage IA, are 85.8% and 80.4%, respectively [3]. Therefore, the detection of small PDAC is very important, but diagnosis using imaging alone is limited [28]. It has been reported that the sensitivity of EUS-FNA in diagnosing malignant pancreatic tumors with a diameter of 10 mm is 73.1%, and a multivariate analysis found that tumor diameter is an independent factor affecting accuracy [11]. In addition, reports indicated that complications associated with EUS-FNA are more likely to occur in small tumors [29]. On the other hand, Ikemoto et al. [16] reported that 86% (31/36) and 77% (30/39) of patients with stage 0 and IA PDAC, respectively, had MPD stenosis on MRI. Therefore, repeat PJC using an ENPD catheter is considered a good indication for patients with suspected PDAC with a small diameter or no mass, regardless of the lesion site.

However, there was no significant difference between the sensitivities of the PJC and ENPD catheter diameters and brush cytology. In a randomized controlled trial comparing the 4-Fr and 5-Fr ENPD catheters, Mouri et al. [30] reported no significant difference in the collection rate of adequate samples for PJC. Moreover, the incidence of post-ERCP pancreatitis was significantly higher with 5-Fr ENPD catheters. This suggests that 4-Fr ENPD catheters may be more suitable for repeat PJC. Yamaguchi et al. [31] reported that brush cytology combined with PJC after brushing significantly increased the diagnostic sensitivity of PDAC. In contrast, Mie et al. [32] retrospectively examined 34 patients who underwent SPACE with brush cytology for pancreatic duct stenosis and reported that the efficacy of adding brush cytology to SPACE was limited. Further prospective studies are required to clarify the effect of brush cytology on the diagnostic yield of PDAC after repeated PJC using an ENPD catheter.

In this study, the cytological diagnosis was atypical in 28% (23/82) of patients, of which 73.9% (17/23) were ultimately diagnosed as PDAC. Although the diagnostic category differs from this study, the World Health Organization (WHO) Reporting System for Pancreaticobiliary Cytopathology [33] describes the estimated risk of malignancy in the “atypical” category as 30–40% for EUS-FNA biopsy and 25–77% for bile duct brushing specimens. As for PJC, patients determined as “atypical” require additional pathological diagnosis or careful follow-up, keeping in mind the possibility of malignancy.

Acute pancreatitis is the most common complication of ERCP and may be fatal in severe cases. Reports indicated the incidence of acute pancreatitis associated with ERCP-related diagnosis of PDAC to be 4.3–11.5% [15,25,34]. In addition, the incidence of acute pancreatitis in patients in whom an ENPD catheter was placed to diagnose PDAC has been reported to be 5.1–7.5%, similar to that of our study (7.3%). Kawamura et al. [15] reported that, in patients with PDAC in whom ERCP-based cytology was performed for diagnosis, the incidence of acute pancreatitis was significantly lower in patients undergoing ENPD catheter placement than those without an ENPD catheter. By placing the tip of the ENPD catheter upstream of the pancreatic duct stenosis, stagnant pancreatic fluid is drained, which may suppress the occurrence of pancreatitis. However, if the tip of the catheter is located within the stenosis, pancreatitis may occur, owing to poor drainage of the pancreatic juice. Therefore, the amount of drained pancreatic juice should be carefully monitored after ENPD catheter placement.

This study has some limitations. First, the number of patients enrolled in this study was insufficient, especially for early-stage PDAC with a tumor diameter of 10 mm or less, which was very small (four patients). Repeated PJC using an ENPD catheter is useful for diagnosing such cases; however, further accumulation of a larger number of cases is necessary to ensure its usefulness. Second, in this study, all cytological evaluations were performed with Papanicolaou stains only. When the evaluation is difficult, especially when the diagnosis is “atypical”, it is necessary to consider immunohistochemical stains such as mutant p53 expression and p16 loss, which are recommended by WHO Reporting System for Pancreaticobiliary Cytopathology [33].

## 5. Conclusions

This prospective study revealed the cumulative effect of up to six repeated PJCs using an ENPD catheter for the diagnosis of PDAC. Especially in small PDAC of 10 mm or less, it exhibited a very good diagnostic yield and is considered a good indication in these cases. However, acute pancreatitis occurs frequently, and careful observation after ENPD catheter placement is essential.

## Figures and Tables

**Figure 1 diagnostics-13-02696-f001:**
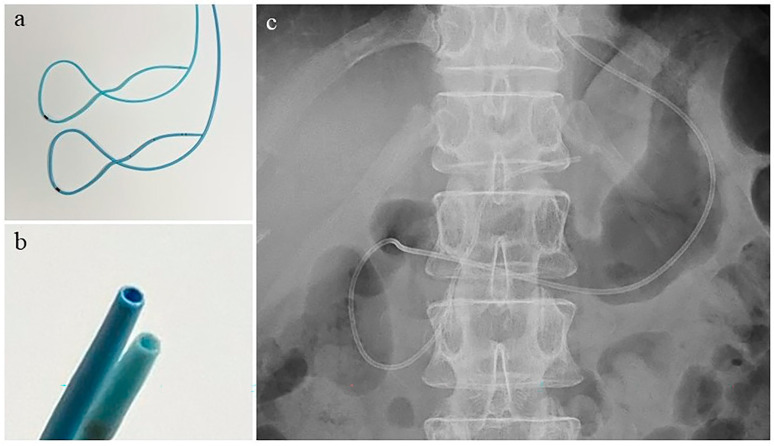
Endoscopic nasopancreatic drainage catheters and placement. (**a**) The 4-Fr (upper) and 5-Fr (lower) catheters have the same looped configuration; (**b**) Cross-sectional view: upper, 5-Fr; lower, 4-Fr; (**c**) The catheter was subsequently placed in the main pancreatic duct. The tip is located in the pancreatic body.

**Figure 2 diagnostics-13-02696-f002:**
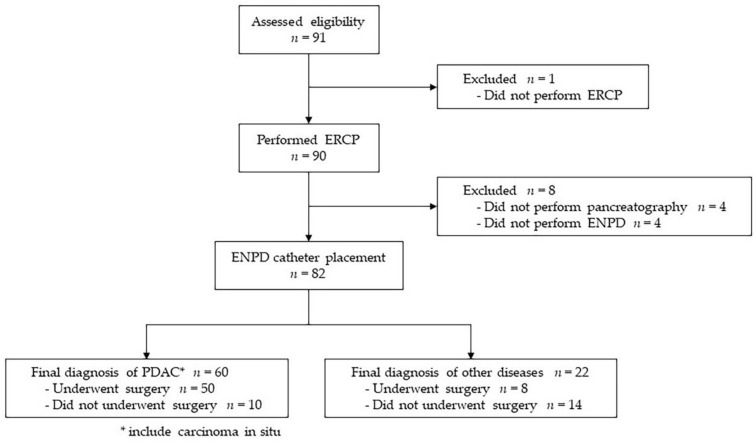
Study flow chart.

**Figure 3 diagnostics-13-02696-f003:**
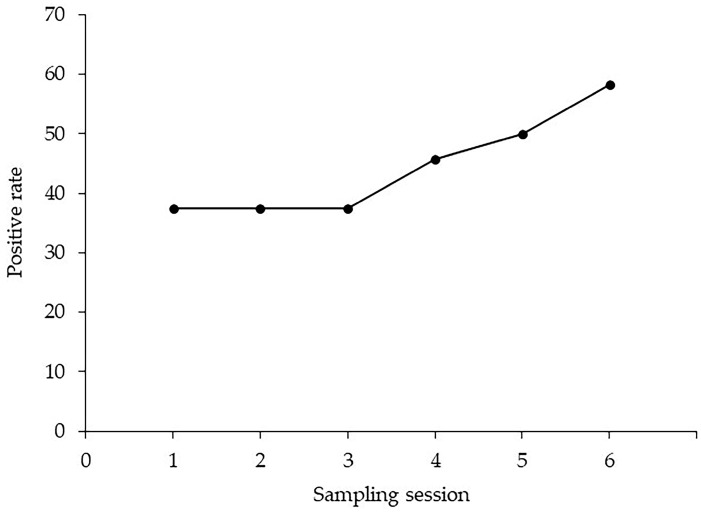
The cumulative positive rate of repeated pancreatic juice cytology.

**Table 1 diagnostics-13-02696-t001:** Clinical characteristics of 82 patients with an ENPD catheter placement.

	Values
Age (years)	72 (64–79)
Sex (male to female)	50:32
Indication for pancreatic juice cytology, *n* (%)	
Pancreatic mass	60 (73.2%)
Pancreatic duct stenosis	13 (15.9%)
Pancreatic duct dilatation	6 (7.3%)
Pancreatic cyst	3 (3.7%)
Diameter of the pancreatic mass on imaging, *n* (%)	
≤10 mm	9 (15.5%)
11–20 mm	27 (46.6%)
>20 mm	22 (37.9%)
Location of the lesion	
Head	38 (46.3%)
Body	25 (30.5%)
Tail	19 (23.2%)
Diameter of the ENPD catheter, *n* (%)	
4-Fr	62 (75.6%)
5-Fr	20 (24.4%)
Unexpected ENPD catheter displacement, *n* (%)	3 (3.7%)
Due to complications	2 (2.4%)
Self-removal	1 (1.2%)
Number of PJC submissions via an ENPD catheter	
Surgical resection performed, *n* (%)	58 (70.7%)
Final diagnosis, *n* (%)	
Pancreatic ductal adenocarcinoma	58 (70.7%)
Carcinoma in situ	2 (2.4%)
Intraductal papillary mucinous neoplasm	2 (2.4%)
Chronic pancreatitis	11 (13.4%)
Autoimmune pancreatitis	1 (1.2%)
Pancreatic cyst	5 (6.1%)
Others	3 (3.7%)

Data are expressed as number (percentage) or median (interquartile range). ENPD, endoscopic nasopancreatic drainage; PJC, pancreatic juice cytology.

**Table 2 diagnostics-13-02696-t002:** Comparison of cytological diagnosis and final diagnosis.

	Final Diagnosis by Histopathological Examination or Clinical Follow-Up		
Cytological Diagnosis	PDAC, *n* (%)	Non-PDAC (Benign), *n* (%)	Total, *n* (%)
Positive	28 (96.6%)	1 (3.4%)	29 (35.4%)
Atypical	17 (73.9%)	6 (26.1%)	23 (28.0%)
Negative	15 (50.0%)	15 (50.0%)	30 (36.6%)
Total, *n* (%)	60 (73.2%)	22 (26.8%)	82 (100)

PDAC, pancreatic ductal adenocarcinoma.

**Table 3 diagnostics-13-02696-t003:** Diagnostic yields of repeated PJC via an ENPD catheter.

	Values
Sensitivity	46.7% (28/60)
Specificity	95.5% (21/22)
Positive predictive value	96.6% (28/29)
Negative predictive value	39.6% (21/53)
Accuracy	59.8% (49/82)

ENPD, endoscopic nasopancreatic drainage; PJC, pancreatic juice cytology.

**Table 4 diagnostics-13-02696-t004:** The positivity rate of pancreatic juice cytology per session.

Session 1	Session 2	Session 3	Session 4	Session 5	Session 6
31.7% (19/60)	25.9% (14/54)	20.0% (10/50)	35.0% (14/40)	37.1% (13/35)	45.8% (11/24)

**Table 5 diagnostics-13-02696-t005:** The diagnostic yield, based on tumor factors and diagnostic procedures.

	Number of Patients	Number of Positivity	Sensitivity	*p* Value
Tumor location				
Head	29	17	58.6%	0.045
Body	19	10	52.6%	
Tail	12	2	16.7%	
Tumor size				0.073
≤10 mm	4	4	100%	
11–20 mm	28	14	50.0%	
>20 mm	28	11	39.3%	
MPD stenosis				0.032
Present	46	26	56.5%	
Absent	14	3	21.4%	
Catheter diameter				0.500
4-Fr	50	23	46.0%	
5-Fr	10	6	60.0%	
Brush cytology				0.511
Performed	19	8	42.1%	
Not performed	41	21	51.2%	

MPD, main pancreatic duct.

**Table 6 diagnostics-13-02696-t006:** Complications.

	All Patients (*n* = 82)	PDAC (*n* = 60)
Pancreatitis	6 (7.3%)	4 (6.7%)
Mild	3 (3.7%)	3 (5.0%)
Moderate	2 (2.4%)	0
Severe	1 (1.2%)	1 (1.7%)
Hyperamylasemia	44 (53.7%)	36 (60.0%)
Cholangitis	0	0

PDAC, pancreatic ductal adenocarcinoma.

## Data Availability

The data presented in this study are available on request from the corresponding author. The data are not publicly available due to privacy.

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
