# Peer review of "Diagnostic Ability and Safety of Repeated Pancreatic Juice Cytology Using an Endoscopic Nasopancreatic Drainage Catheter for Pancreatic Ductal Adenocarcinoma: A Multicenter Prospective Study"

_diagnostics, 2023, doi:10.3390/diagnostics13162696_

Round 1

Reviewer 1 Report

Manuscript entitled " Diagnostics ability and safety of repeated pancratc juice cytology using an endoscopic nasopancreatic drainage catheter for pancreatic ductal adenocarcinoma : a multicenter prospctive study."

This is an interesting study showing the usefulness of endoscopic drainage of pancreatic juice for diagnosis of pancreatic ductal adenocarcinoma (PDAC). Patients enrolled in the study were suspected to pancreatic surgery and selected basing on preliminary imaging diagnosis by CT, MRI and EUS. Six samples of juice was collected from each patients during 48 hours. Cytological examination of pancreatic juice obtained as the result of nasopancreatic drainage supports the diagnosis with sensitivity of 46,7% and specificity 95.5%.The highest sensitivity concerns the tumors of pancreatic head. As complications were noted pancreatitis mostly mild, (single was severe), and  hyperamylasemia. Authors concluded that used procedure appears as good diagnostic  yield, especially for a small PDAC.

Study was carefully planned and performed and seems that used method of diagnosis PDC could be applicable for  clinical use.

Comments :

Page 11, line 349-345  The sentence is unfinished.

References :    Ref no 30 correction of citation

This is an interesting study showing the usefulness of endoscopic drainage of pancreatic juice for diagnosis of pancreatic ductal adenocarcinoma (PDAC). Patients enrolled in the study were suspected to pancreatic surgery and selected basing on preliminary imaging diagnosis by CT, MRI and EUS. Six samples of juice was collected from each patients during 48 hours. Cytological examination of pancreatic juice obtained as the result of nasopancreatic drainage supports the diagnosis with sensitivity of 46,7% and specificity 95.5%.The highest sensitivity concerns the tumors of pancreatic head. As complications were noted pancreatitis mostly mild, (single was severe), and  hyperamylasemia. Authors concluded that used procedure appears as good diagnostic  yield, especially for a small PDAC.

Comments :

Page 11, line 349-345  The sentence is unfinished.

References :    Ref no 30 correction of citation

Reviewer 2 Report

The review pertains to the article entitled Diagnostic ability and safety of repeated pancreatic juice cytol-2 ogy using an endoscopic nasopancreatic drainage catheter for 3 pancreatic ductal adenocarcinoma: A multicenter prospective 4 study 5 submitted for Diagnostics Journal.

The present article is in my mind a follow up of the efforts of numerous hospitals in Japan to better diagnose pancreatic ductal adenocarcinoma. Judging from the resources and a quick PubMed search, it has been a major concern for some of the authors. 

The study design was a multicenter, open-label, uncontrolled study conducted at Hiroshima University Hospital and 12 affiliated hospitals. The study enrolled patients suspected of having resectable PDAC based on imaging studies and required cytology. The primary endpoint was the sensitivity of repeat PJC using an ENPD catheter, while the secondary endpoint was the incidence of complications associated with catheter placement and removal. The results of the study showed that the cumulative effect of up to six repeated PJC sessions using an ENPD catheter resulted in a positive rate of 58.3% for the diagnosis of PDAC. The sensitivity of a single PJC session was low (31.7%), but it increased with the number of sessions. The diagnostic performance was influenced by tumor factors such as location and size, with higher sensitivity observed in tumors located in the pancreatic head and smaller tumors (≤10 mm). Complications, particularly acute pancreatitis, occurred in a small percentage of patients. The study has some limitations, including the relatively small number of patients enrolled, especially for early-stage PDAC. Further studies with a larger sample size are needed to confirm the usefulness of repeated PJC in these cases.

In conclusion, this prospective study highlights the cumulative effect of repeated PJC using an ENPD catheter for the diagnosis of PDAC. It demonstrates higher sensitivity with increased sessions and suggests that repeated PJC is particularly beneficial for diagnosing small PDAC. However, cautious monitoring for complications, especially acute pancreatitis, is essential when using the ENPD catheter.

The overall article is well written, the statistical analysis is well presented although I would like to see who handled the analysis (not stated in author contribution).

Sample size is small as stated before but it’s not a major drawback of the article. Overall, the present article merits publication.

Can be published as is.

Author Response

RESPONSE TO REVIEWER 2:

We wish to express our appreciation to the Reviewer for the insightful comments, which have helped us significantly improve the paper.

Comment 1: I would like to see who handled the analysis (not stated in author contribution).

Response: We thank the Reviewer for this pertinent comment. In accordance with the Reviewer’s comment, we have added the following text in the Author Contributions:

“formal analysis, S.N. and Y.I.;”

Thank you again for your comments on our paper. We trust that the revised manuscript is suitable for publication.

Reviewer 3 Report

I thank the Editor for having submitted this interesting paper to me.

The aim of the study is well defined and of current importance and the design well articulated.

However, revisions on diagnostic cytological categories and data presentation are needed.

First: the 'false positive' diagnostic category is not contemplated in a prospective study because it refers to data that can be acquired only in a follow-up (surgical or clinical).

Second: the results and sensivity and specifity can be evaluated by comparing the three diagnostic cyitological categories (positive, doubtful and negative) with the definitive data (surgical or clinical; see attached table). This is the important point of the study.

Results, Discussion and Conclusion must be re-elaborated on these data. Finally, it is necessary to underline, among the limitations, the inadequacy of traditional cytology for the immunohistochemistry method aimed at discriminating the histotypes according to WHO (doi: 10.1111/1759-7714.14581).

Positive surgery

Negative surgery/Other

p Value/statistics

Positive cytology

Doubt cytology

Negative cytology

Total cases

Author Response

RESPONSE TO REVIEWER 3:

We wish to express our appreciation to the Reviewer for the insightful comments, which have helped us significantly improve the paper.

Comment 1: I would like to see who handled the analysis (not stated in author contribution). The 'false positive' diagnostic category is not contemplated in a prospective study because it refers to data that can be acquired only in a follow-up (surgical or clinical).

Response: We thank the Reviewer for this pertinent comment. In accordance with the Reviewer’s comment, we have changed “false positive (atypical cells of unknown malignancy)” to “suspicious (difficult to distinguish between benign and malignant)”.

Comment 2: The results and sensitivity and specificity can be evaluated by comparing the three diagnostic cytological categories (positive, doubtful and negative) with the definitive data (surgical or clinical; see attached table). This is the important point of the study.

Response: We appreciate the Reviewer’s comment. In accordance with the Reviewer’s comment, we have added the following text in the Results (p. 8, lines 242-252).

“Twenty-nine patients had positive PJC, and 23 underwent surgical resection. Finally, 28 of 29 patients (96.6%) were diagnosed with PDAC. Twenty-three patients had suspicious PJC. Of the 23 patients, 11 underwent EUS-FNA and eight were determined to be positive (ma-lignant). Seventeen patients underwent surgical resection and six were followed up with imaging. Finally, 17 of 23 patients (73.9%) were diagnosed with PDAC. Thirty patients had negative PJC. EUS-FNA was performed in 12 patients, and three were determined to be positive (malignant). On the other hand, seven of the nine patients who were deter-mined to be negative on both PJC and EUS-FNA were difficult to differentiate from PDAC and underwent surgery, and five were diagnosed with PDAC. Finally, 15 of 30 patients (50.0%) were diagnosed with PDAC.”

Comment 3: It is necessary to underline, among the limitations, the inadequacy of traditional cytology for the immunohistochemistry method aimed at discriminating the histotypes according to WHO (doi: 10.1111/1759-7714.14581).

Response: We appreciate the Reviewer’s comment. Unlike lung cancer, the necessity and usefulness of immunohistochemical staining in cytological and histological examination of PDAC in limited, and it is rarely performed in routine clinical practice. Therefore, we do not believe that it is necessary to include the content the Reviewer pointed out in the limitation.

Thank you again for your comments on our paper. We trust that the revised manuscript is suitable for publication.

Round 2

Reviewer 3 Report

I congratulate the Authors for the improvements made to the paper.

Unfortunately, I seem to remember reporting other changes needed for a correct study description that I don't find in the new version of the text.

I hope that the Authors wish and trust, first of all, to develop a scientifically correct paper*.

No more words.

Author Response

RESPONSE TO REVIEWER 3:

We wish to express our appreciation to the Reviewer for the insightful comments, which have helped us significantly improve the paper.

We have carefully reconsidered your comments and revised our manuscript.

Comment 1: I would like to see who handled the analysis (not stated in author contribution). The 'false positive' diagnostic category is not contemplated in a prospective study because it refers to data that can be acquired only in a follow-up (surgical or clinical).

Response: We thank the Reviewer for this pertinent comment. In accordance with the Reviewer’s comment, we have changed “false positive (atypical cells of unknown malignancy)” to “atypical (atypical cells of unknown malignancy)”.

Comment 2: The results and sensitivity and specificity can be evaluated by comparing the three diagnostic cytological categories (positive, doubtful and negative) with the definitive data (surgical or clinical; see attached table). This is the important point of the study.

Response: We appreciate the Reviewer’s comment. In accordance with the Reviewer’s comment, we have added the following text in the Results (p. 8, lines 242-248).

“Table 2 shows the final diagnosis based on histopathological examination of specimens obtained by surgical resection or EUS-FNA, or clinical follow-up for each cytological diagnosis. Twenty-nine patients (35.4%) had positive PJC, and 28 patients (96.6%) were diagnosed with PDAC. Twenty-three patients (28.0%) had atypical PJC, and 17 patients (73.9%) were diagnosed with PDAC. Thirty patients (36.6%) had negative PJC, but 15 patients (50.0%) were diagnosed with PDAC.”

we have added Table 2 comparing the three cytological diagnostic categories and the final diagnosis.

Table 2. Comparison of cytological diagnosis and final diagnosis

Final diagnosis by histopathological examination or clinical follow-up

Cytological diagnosis

PDAC, n (%)

Non-PDAC (benign), n (%)

Total, n (%)

Positive

28 (96.6%)

1 (3.4%)

29 (35.4%)

Atypical

17 (73.9%)

6 (26.1%)

23 (28.0%)

Negative

15 (50.0%)

15 (50.0%)

30 (36.6%)

Total, n (%)

60 (73.2%)

22 (26.8%)

82 (100)

PDAC, pancreatic ductal adenocarcinoma

In addition, we have added the following text in the Discussion (p. 12, lines 356-362).

“In this study, the cytological diagnosis was atypical in 28% (23/82) of patient, of which 73.9% (17/23) were ultimately diagnosed as PDAC. Although the diagnostic category differs from this study, the World Health Organization (WHO) Reporting System for Pancreaticobiliary Cytopathology [33] describes the estimated risk of malignancy in the “atypical” category as 30–40% for EUS-FNA biopsy and 25–77% for bile duct brushing specimens. As for PJC, patients determined as “atypical” require additional pathological diagnosis or careful follow-up, keeping in mind the possibility of malignancy.”

Comment 3: It is necessary to underline, among the limitations, the inadequacy of traditional cytology for the immunohistochemistry method aimed at discriminating the histotypes according to WHO (doi: 10.1111/1759-7714.14581).

Response: We appreciate the Reviewer’s comment. In accordance with the Reviewer’s comment, we have added the following text in the Discussion (p. 12, lines 380-384)

“Second, in this study, all cytological evaluation were performed with Papanicolaou stains only. When the evaluation is difficult, especially when the diagnosis is “atypical”, it is necessary to consider immunohistochemical stains such as mutant p53 expression and p16 loss, which are recommended by WHO Reporting System for Pancreaticobiliary Cytopathology [33].”

Thank you again for your comments on our paper. We trust that the revised manuscript is suitable for publication.

Round 3

Reviewer 3 Report

The paper has been improved.

no more changes.